

# Light-induced topological magnons
# in two-dimensional van der Waals magnets

Emil Viñas Boström[1*], Martin Claassen[2], James W. McIver[1],
Gregor Jotzu[1], Angel Rubio[1,2,†] and Michael A. Sentef[1,3,‡]

**1** Max Planck Institute for the Structure and Dynamics of Matter,
Luruper Chaussee 149, 22761 Hamburg, Germany
**2** Center for Computational Quantum Physics, The Flatiron Institute,
162 Fifth Avenue, New York, NY 10010, United States of America
**3** Institute for Theoretical Physics, University of Bremen,
Otto-Hahn-Allee 1, 28359 Bremen, Germany

★ emil.vinas-bostrom@mpsd.mpg.de, † angel.rubio@mpsd.mpg.de, ‡ michael.sentef@mpsd.mpg.de

## Abstract

Driving a two-dimensional Mott insulator with circularly polarized light breaks time-reversal and inversion symmetry, which induces an optically-tunable synthetic scalar spin chirality interaction in the effective low-energy spin Hamiltonian. Here, we show that this mechanism can stabilize topological magnon excitations in honeycomb ferromagnets and in optical lattices. We find that the irradiated quantum magnet is described by a Haldane model for magnons that hosts topologically-protected edge modes. We study the evolution of the magnon spectrum in the Floquet regime and via time propagation of the magnon Hamiltonian for a slowly varying pulse envelope. Compared to similar but conceptually distinct driving schemes based on the Aharanov-Casher effect, the dimensionless light-matter coupling parameter $\lambda = eEa/\hbar\omega$ at fixed electric field strength is enhanced by a factor $\sim 10^5$. This increase of the coupling parameter allows to induce a topological gap of the order of $\Delta \approx 2$ meV with realistic laser pulses, bringing an experimental realization of light-induced topological magnon edge states within reach.

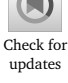

# 1   Introduction

The experimental realization of magnetic van der Waals (vdW) materials with a thickness down to the monolayer limit has sparked a new interest in fundamental aspects of two-dimensional magnetism [1–4]. Due to a competition of strong anisotropy, fluctuations, and spin-orbit effects, two-dimensional vdW materials are known to exhibit diverse magnetic orders ranging between semiconducting ferromagnetism, itinerant ferromagnetism, and insulating antiferromagnetism [5–8]. However, these properties also make them prime candidates to host topological phenomena such as Berezinskii-Kosterlitz-Thouless phase transitions [9], quantum spin liquids [10,11], magnetic skyrmions [12], and fractional excitations [13].

In addition to the intrinsic topological properties of vdW magnets, the tremendous progress in functionalization of materials through light-matter coupling [14–21] shows that it is possible to manipulate the magnetic and topological order of such materials using laser fields. In recent theoretical studies it has been shown that driving a two-dimensional Mott insulator with circularly polarized light breaks both time-reversal and inversion symmetries. This is reflected by an induced scalar spin chirality interaction that governs the transient dynamics of low-energy spin excitations [11,22]. Remarkably, optical irradiation red-detuned from the Mott gap can limit heating and absorption to enable a controlled realization of such Floquet-engineered spin dynamics, and it has been argued for a Kagomé lattice antiferromagnet that the spin chirality term leads to a chiral spin liquid ground state in herbertsmithite and kapellasite [11]. Experimental realizations of Floquet-engineered spin Hamiltonians have also been demonstrated for both classical [23] and quantum magnetism [24] using ultracold atoms in driven optical lattices [25].

In this work, we demonstrate that the photo-induced scalar spin chirality has consequences for the low-energy magnetic excitations of ferromagnetic systems. In particular, for honeycomb ferromagnets it leads to a magnon Haldane model [26] with a topological gap and chiral magnon edge states [27]. To this end, we first derive the magnitude of the induced time-reversal symmetry breaking contribution for a honeycomb Mott insulator. We then show that application of the effective spin Hamiltonian, with parameters taken from the prototypical monolayer vdW magnet $CrI_3$ [2, 28–30], can lead to a gap $\Delta \approx 2$ meV in the magnon spectrum for a realistic field strength $E = 10^9$ V/m and photon energy $\hbar\omega = 1$ eV, inducing non-zero Chern numbers and leading to chiral magnon edge states. Importantly, we find that

the dimensionless Floquet parameter that describes the magnitude of light-matter interaction is enhanced by a factor $\sim 10^5$ compared to similar but conceptually distinct driving schemes based on the Aharanov-Casher effect for pure spin models [27, 31], since the electric field couples to the charge instead of the magnetic moment. This amplification is shown to be crucial for a potential experimental realization of a topological magnon phase in monolayer vdW magnets.

## 2 Model

To assess the magnitude of photo-induced time-reversal symmetry breaking for honeycomb Mott insulators, we commence by deriving an effective transient spin-1/2 Hamiltonian from a single-band Mott insulator

$$H = -t \sum_{\langle ij \rangle \sigma} e^{i\theta_{ij}(\tau)} c_{i\sigma}^{\dagger} c_{j\sigma} + U_0 \sum_i \hat{n}_{i\uparrow} \hat{n}_{i\downarrow} + \frac{V}{2} \sum_{\langle ij \rangle} \hat{n}_i \hat{n}_j - J_D \sum_{\langle ij \rangle} \hat{\mathbf{S}}_i \cdot \hat{\mathbf{S}}_j, \qquad (1)$$

where $c_{i\sigma}^{\dagger}$ creates an electron at site $i$ with spin projection $\sigma$, $\hat{n}_{i\sigma} = c_{i\sigma}^{\dagger} c_{i\sigma}$ is the local spin density, $\hat{n}_i = \hat{n}_{i\uparrow} + \hat{n}_{i\downarrow}$ is the local electron density, $t$ is the hopping amplitude between nearest neighbor sites $i$ and $j$, and $U_0$ is a local interaction. We also consider nearest neighbor direct and exchange interactions $V$ and $J_D$, the later being expressed in terms of the spin operator $\hat{\mathbf{S}}_i = c_{i\sigma}^{\dagger} \boldsymbol{\sigma}_{\sigma\sigma'} c_{i\sigma'}$ where $\boldsymbol{\sigma}$ is the vector of Pauli matrices [1]. We use the Einstein summation convention for repeated spin indexes.

The electrons interact with an external electromagnetic field described via the Peierls phases

$$\theta_{ij}(\tau) = -\frac{e}{\hbar} \int_{\mathbf{r}_j}^{\mathbf{r}_i} d\mathbf{r} \cdot \mathbf{A}(\mathbf{r}, \tau). \qquad (2)$$

To break time-reversal symmetry and induce a scalar spin chirality, we use a circularly polarized laser with vector potential $\partial_t \mathbf{A}(\mathbf{r}, \tau) = -E(\tau)(\cos \omega \tau, \zeta \sin \omega \tau)$ in the dipole approximation, where $\zeta = \pm 1$ for right/left-handed polarization. Assuming a constant envelope $E(\tau) = E$ and writing $\boldsymbol{\delta}_{ij} = \mathbf{r}_i - \mathbf{r}_j = a(\cos \phi_{ij}, \sin \phi_{ij})$ with $a$ the lattice constant, the Peierls phases are $\theta_{ij}(\tau) = -\lambda \sin(\omega \tau - \zeta \phi_{ij})$. The dimensionless quantity $\lambda = eEa/\hbar\omega$ determines the effective field strength of the laser. In an optical lattice, $eE$ is replaced by the driving force $F$, which may result from an acceleration of the lattice [32] or a magnetic field gradient [33].

Although the above model provides a simplified description of realistic monolayer vdW magnets, neglecting both the multi-orbital structure of the transition metals ions and the superexchange processes induced by interactions with the surrounding halides [34, 35], it provides a starting point for more advanced treatments. Further, since the topological properties of honeycomb ferromagnets are determined by the lattice structure and the presence or absence of time-reversal symmetry [26], we expect the model to give a correct description of the topological features of the magnon excitations.

## 3 Effective spin Hamiltonian

We now construct an effective spin Hamiltonian for driving frequencies $J \ll \hbar\omega \ll U$, where $J \sim t^2/U$ is the leading order Heisenberg exchange in equilibrium. We have followed the

---

[1] The Heisenberg term arises by writing the exchange interaction as $c_{i\sigma}^{\dagger} c_{i\sigma} c_{j\sigma'}^{\dagger} c_{j\sigma'} = \frac{1}{2} \hat{n}_i \hat{n}_j + 2\hat{\mathbf{S}}_i \cdot \hat{\mathbf{S}}_j$, and absorbing the first term into the direct interaction by a renormalization of $V$.

method of Ref. [11] to obtain the effective Hamiltonian to fourth order in $t/U$ for a periodic external field. For a slowly varying envelope $E(\tau)$ the Hamiltonian is almost periodic with the period $H(\tau + 2\pi/\omega) = H(\tau)$. This allows us to employ Floquet theory and rewrite the electronic Hamiltonian exactly using a Fourier expansion

$$H = -t\sum_{\langle ij \rangle \sigma}\sum_{mm'}\mathcal{J}_{m-m'}(\lambda)e^{i(m-m')\zeta\phi_{ij}}c_{i\sigma}^{\dagger}c_{j\sigma}\otimes|m\rangle\langle m'| + H_I\otimes\mathbf{1} - \sum_m m\omega\otimes|m\rangle\langle m|, \quad (3)$$

expressed in the product space of the electronic Hamiltonian and the space of periodic functions [36] denoted by Fourier modes $|m\rangle$, which can be identified with the classical limit of $m$ absorbed or emitted virtual photons. Here, $\mathcal{J}_m(x)$ is the Bessel function of the first kind of order $m$. The interaction Hamiltonian is $H_I = U\sum_i \hat{n}_{i\uparrow}\hat{n}_{i\downarrow} - J_D\sum_{\langle ij \rangle}\hat{\mathbf{S}}_i\cdot\hat{\mathbf{S}}_j$, where the nearest neighbor direct interaction has been absorbed by a renormalization of the Hubbard $U$ [37]. Using quasi-degenerate perturbation theory to simultaneously integrate out the doubly occupied states and the $m\neq 0$ Floquet states [11, 38, 39], the effective honeycomb lattice spin Hamiltonian corresponding to the electronic system is given to fourth order in $t/U$ by

$$\mathcal{H} = \sum_{\langle ij \rangle}J_{ij}\hat{\mathbf{S}}_i\cdot\hat{\mathbf{S}}_j + \sum_{\langle\langle ik \rangle\rangle}J'_{ik}\hat{\mathbf{S}}_i\cdot\hat{\mathbf{S}}_k + \sum_{\langle\langle ik \rangle\rangle}\chi_{ik}\hat{\mathbf{S}}_j\cdot(\hat{\mathbf{S}}_i\times\hat{\mathbf{S}}_k). \quad (4)$$

Here $J$ and $J'$ are respectively the nearest and next-nearest neighbor light-induced Heisenberg exchanges, and $\chi$ is a synthetic scalar spin chirality. A non-zero value of $\chi$ signals a non-coplanar spin texture and can appear in equilibrium due to e.g. Dzyaloshinskii-Moriya interactions or geometric frustration [40, 41]. For electrons hopping around closed loops in such a spin texture the spin chirality acts as an effective magnetic field that can give rise to the topological Hall effect [42]. The full expressions for the spin parameters are given in Appendix A.

We note that $J$ has contributions from all even orders in $t/U$, while $J'$ and $\chi$ appear only at fourth order. On the honeycomb lattice, a non-zero spin chirality arises due hopping processes that enclose an isosceles triangle within the hexagons, as indicated schematically in Fig. 1a (and discussed further in Appendix B). Such processes lead to a net phase accumulation in analogy with electrons moving in closed loops in an external magnetic field, and lead to time-reversal symmetry breaking. However, in contrast to using an external magnetic field, driving with a circularly polarized electric field conserves the $SU(2)$ spin symmetry. In the non-interacting limit the corresponding complex next-nearest neighbor tunneling has already been implemented in optical lattices using circular driving [43].

# 4 Antiferromagnetic systems

In the following we assume $|\chi|\ll|J|$, so that depending on the sign of $J$ the system is either ferromagnetic ($J < 0$) or antiferromagnetic ($J > 0$). It has previously been shown that topological magnon edge states can be induced by a constant electric field gradient that splits the magnon bands into Landau levels and leads to a magnon version of the quantum (spin) Hall effect in (anti-) ferromagnets [44, 45]. In the present work the homogeneous but time-dependent electric field instead opens a gap at magnon band crossings, leading to a magnon analog of the quantum anomalous Hall effect. In the antiferromagnetic regime we find that the bands are nearly degenerate with no crossings, and the system remains in a topologically trivial phase. This agrees with previous work where the edge modes of the Néel state were shown to be topologically trivial [46, 47]. However, by adding an in-plane magnetic field [46] or considering an antiferromagnetically coupled bilayer [48], topological magnon edge states

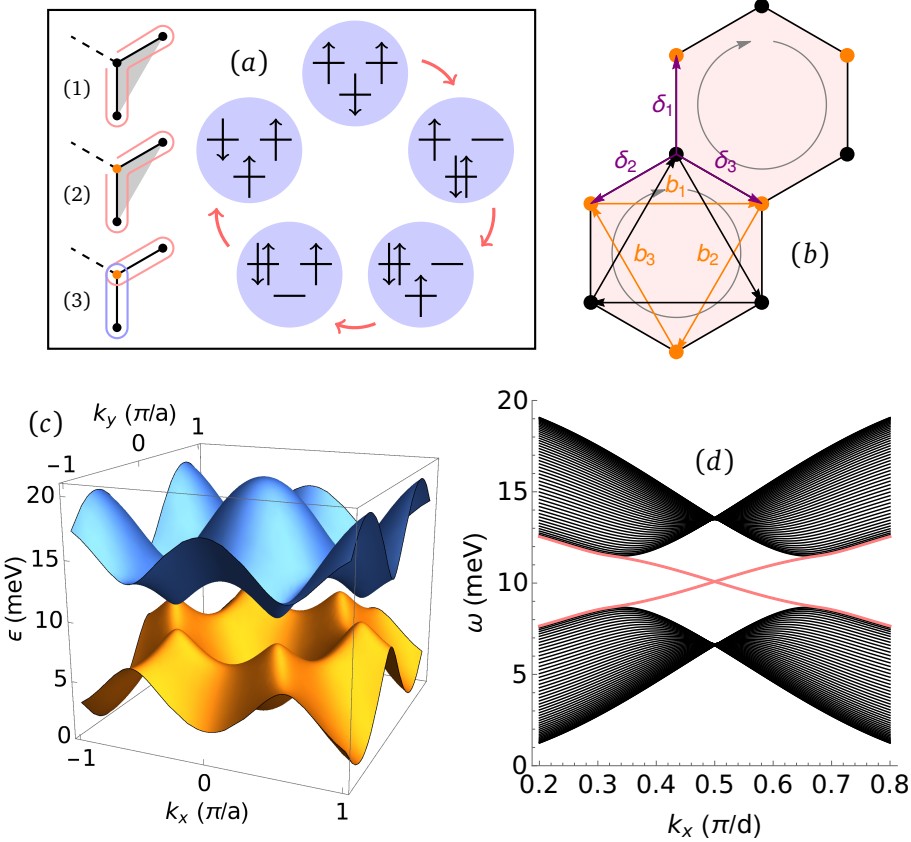

Figure 1: **Chiral light-induced topology.** ($a$) Illustration of the different fourth-order hopping processes available on the honeycomb lattice: (1) a process where the second intermediate state contains a holon-doublon pair at non-zero Floquet index. (2) A process where the system returns to half-filling in the second step (at the site indicated in orange) at non-zero Floquet index. (3) A process where the system returns to half-filling and zero Floquet index in the second step (at the site indicated in orange). The right panel gives an example of a process in class (1) that breaks time-reversal symmetry and induces a scalar spin chirality. ($b$) Portion of the honeycomb lattice illustrating the lattice vectors $\mathbf{b}_i$, colored in black or orange depending on the sublattice, and the nearest neighbor lattice vectors $\boldsymbol{\delta}_i$ (purple). The lattice vectors shown correspond to positive Haldane phases ($\nu_{ik} = 1$) arising from hopping in a clockwise direction. ($c$) Topological magnon bands for a ferromagnetic system with $S = 3/2$, $J = 2.27$ meV, $J' = 0.005$ meV and $\chi = 0.13$ meV, giving a light-induced gap of magnitude $\Delta = 6\sqrt{3}\chi S = 2.07$ meV at the Dirac points. ($d$) Magnon bands for a zig-zag ribbon with $n_y = 100$ for the same parameters as in ($c$). The chiral topological edge states are indicated in pink.

can be induced. Although we focus below on the ferromagnetic state, we expect that an application of our formalism to the non-collinear and bilayer antiferromagnetic cases would lead to similar conclusions.

# 5   Magnons on the honeycomb lattice

We denote the lattice vectors of the honeycomb lattice by $\mathbf{b}_i$ and the vectors between nearest neighbor sites by $\boldsymbol{\delta}_i$ (see Fig. 1b). On the honeycomb lattice the angles $\Phi_{ik} = \zeta(\phi_{ij} - \phi_{kj})$ between next-nearest neighbor sites are given by $\Phi_{ik} = 2\pi\zeta\nu_{ik}/3$, where $\nu_{ik} = 1$ ($\nu_{ik} = -1$) for hopping in a clockwise (anti-clockwise) direction (see Fig. 1b). This leads to a spin chirality of the form $\chi_{ik} = \zeta\nu_{ik}\chi$ where the sign alternates depending on the bond direction.

For a ferromagnetic ground state we can solve the system to leading order in $S^{-1}$ using the Holstein-Primakoff transformation $\hat{S}_i^z = S - a_i^\dagger a_i$, $\hat{S}_i^- \approx \sqrt{2S}a_i^\dagger$ and $\hat{S}_i^+ \approx \sqrt{2S}a_i$ for the spins on sublattice $A$, and similarly but with $a_i \to b_i$ for the spins on sublattice $B$. In terms of its Fourier components the Hamiltonian can be written as $\mathcal{H} = \sum_\mathbf{k} \Psi_\mathbf{k}^\dagger \mathcal{H}_\mathbf{k} \Psi_\mathbf{k}$, where $\Psi_\mathbf{k} = (a_\mathbf{k}, b_\mathbf{k})^T$, $\mathcal{H}_\mathbf{k} = h_0\mathbf{1} + \mathbf{h}\cdot\boldsymbol{\sigma}$, $\mathbf{h} = (h_x, h_y, h_z)$ and $\boldsymbol{\sigma}$ is the vector of Pauli matrices. The eigenvalues of this matrix are $\epsilon_\pm(\mathbf{k}) = h_0(\mathbf{k}) \pm \sqrt{\mathbf{h}(\mathbf{k})\cdot\mathbf{h}(\mathbf{k})}$, where $h_0 = 3JS + 6J'S + 2J'S\xi_\mathbf{k}$, $h_x + ih_y = -JS\rho_\mathbf{k}$ and $h_z = 2\zeta\chi S\sigma_\mathbf{k}$. To simplify the notation we have defined the quantities $\rho_\mathbf{k} = \sum_i e^{-i\mathbf{k}\cdot\boldsymbol{\delta}_i}$, $\xi_\mathbf{k} = \sum_i \cos(\mathbf{k}\cdot\mathbf{b}_i)$ and $\sigma_\mathbf{k} = \sum_i \sin(\mathbf{k}\cdot\mathbf{b}_i)$.

In Fig. 1c we show the magnon band structure for equilibrium spin parameters of bulk CrI$_3$ [28]. For $\chi = 0$ the system has Dirac points at $\mathbf{K}_\eta = \eta(4\pi/3\sqrt{3}, 0)$ (where $\eta = \pm$), while for $\chi > 0$ a gap is opened of magnitude $\Delta = 6\sqrt{3}\chi S$. As shown below, the gap opening is associated with a transition to a non-trivial topological state.

## 5.1   Chern numbers and edge states

To determine the topological structure of the system for $\chi > 0$, we calculate the Chern numbers of the dressed magnon bands. Since the dominant contribution to the Berry curvature comes from the regions around the Dirac points, we expand the Hamiltonian around $\mathbf{K}_\eta$. To linear order in $\boldsymbol{\kappa} = \mathbf{k} - \mathbf{K}$ we find the Hamiltonian $\mathcal{H}^\eta = \nu\eta\kappa_x\sigma_x - \nu\kappa_y\sigma_y + w\eta\sigma_z$, where $\nu = 3JS/2$, $w = -3\sqrt{3}\zeta\chi S$, and we have neglected constant terms proportional to $J$ and $J'$.

The Berry potential for the quasi-stationary state is obtained from the expression [49]

$$v_s^\eta(\mathbf{k}) = \text{Im} \frac{\langle\psi_{ks}|\nabla\mathcal{H}^\eta|\psi_{k,-s}\rangle \times \langle\psi_{k,-s}|\nabla\mathcal{H}^\eta|\psi_{ks}\rangle}{(\epsilon_+(\mathbf{k}) - \epsilon_+(\mathbf{k}))^2}, \tag{5}$$

where $|\psi_{ks}\rangle$ are the eigenstates of $\mathcal{H}^\eta$ and $s = \pm$ denotes the upper/lower magnon branch. It is clear from the cross product that $v_{-s} = -v_s$ and so it is sufficient to compute $v_+$. We calculate the matrix elements by noting that $\partial_{\kappa_x}\mathcal{H}^\eta = \nu\eta\sigma_x$ and $\partial_{\kappa_y}\mathcal{H}^\eta = -\nu\sigma_y$, and defining $d^2 = \mathbf{h}\cdot\mathbf{h} = \frac{1}{4}(\epsilon_+(\mathbf{k}) - \epsilon_-(\mathbf{k}))^2$ we find the Berry potential $v_s^\eta = -s\eta\nu^2 h_z/(2d^3)$. Since the Chern number is the integral over the Berry potential we have

$$\mathcal{C}_s = \frac{1}{2\pi}\sum_\eta \int d^2\kappa\, v_s^\eta(\mathbf{k}) = s\zeta\,\text{sgn}(\chi). \tag{6}$$

For positive $\chi$ the upper (lower) band has a Chern number $\mathcal{C} = \zeta$ ($\mathcal{C} = -\zeta$).

The non-zero Chern numbers imply the existence of topological magnon edge states. We verify this explicitly for a ribbon geometry with zig-zag edges, periodic in the $x$-direction and with $n_y$ sites in the $y$-direction. In Fig. 1d we show the band structure of the ribbon for $n_y = 100$, where chiral edge states are situated in the bulk band gap and connect the Dirac

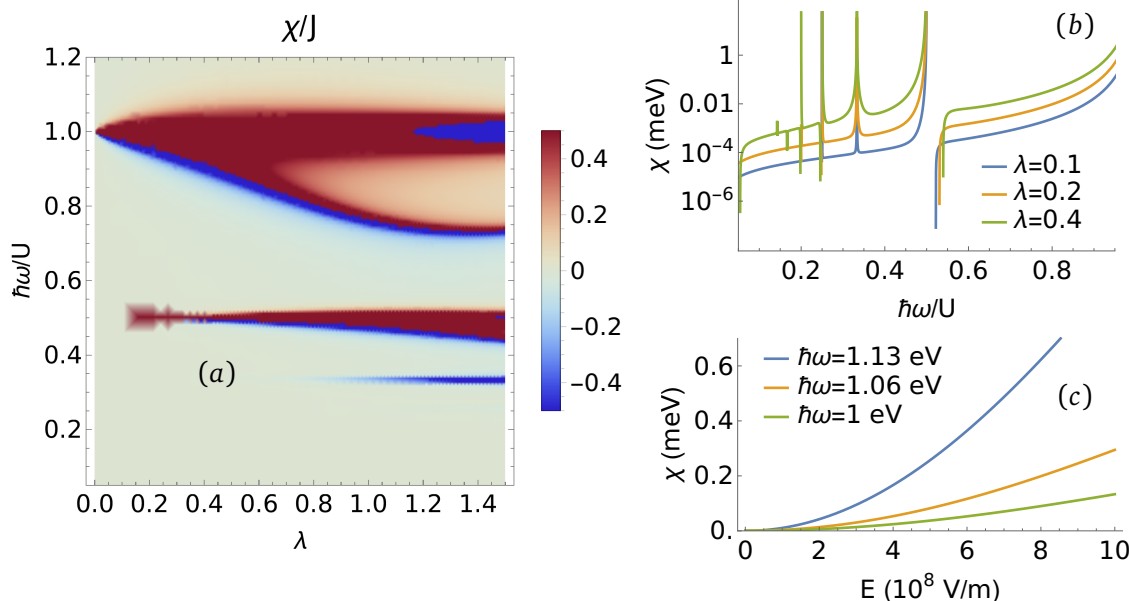

Figure 2: **Energy scales of light-induced chirality.** ($a$) Ratio of the synthetic scalar spin chirality $\chi$ to the field-renormalized Heisenberg exchange $J$ as a function of field strength $\lambda$ and photon energy $\hbar\omega$. The results are for a system with $t = 50$ meV, $U = 1.25$ eV and $J_D = 12$ meV, giving an effective ferromagnetic exchange in equilibrium of $J = 4.0$ meV and $J' = 13$ $\mu$eV. ($b$) $\chi$ as a function of photon energy $\hbar\omega$ for different values of the effective field strength $\lambda$. ($c$) $\chi$ as a function of electric field strength $E$ for different values of the photon energy $\hbar\omega$.

points at $\mathbf{K}_+$ and $\mathbf{K}_-$. We find no qualitative differences between the edge states of a zig-zag and armchair ribbon, and also in the latter case the magnon band structure is topological. However, for an armchair ribbon the $\mathbf{K}$ and $\mathbf{K}'$ points of the two-dimensional Brillouin zone are projected onto the same momentum of the surface Brillouin zone, which could have important consequences when trying to populate the magnon states, since this process is typically restricted to small momentum transfers.

We note that in contrast to the edge states of a quantum spin Hall insulator, the edge magnons of different chirality are located on opposite edges of the sample. Since the sign of the Chern numbers and thereby the chirality of the edge states are determined by the polarization of the optical field, this allows to control the propagation direction of the edge magnons by changing the helicity of the field.

## 6 Parameter dependence of the scalar spin chirality

We have seen that a non-zero value of the scalar spin chirality $\chi$ leads to a topological magnon state. We now discuss the values of the frequency and electric field strength needed to induce this state in a system with the spin parameters of CrI$_3$.

We start by considering the ratio $\chi/J$ as a function of the photon energy $\hbar\omega$ and effective field strength $\lambda$, which is a measure of the ratio between the bandgap and the bandwidth. The results are shown in Fig. 2a for the electronic parameters $t = 50$ meV, $U = 1.25$ eV and $J_D = 12$ meV corresponding approximately to monolayer CrI$_3$ [2]. We find a resonant behavior

---

[2]We have calculated the value of $U$ by DFT+U simulations of monolayer CrI$_3$ using the Octopus TD-DFT code.

in $\chi/J$ when the frequency $\hbar\omega = U/n$ with $n$ integer, which corresponds to the thresholds for $n$-photon excitation across the Mott gap. In addition, the diagonal feature extending across Fig. 2a indicates the transition from a ferromagnetic to an antiferromagnet effective exchange parameter [38]. Approaching this transition while simultaneously ensuring $|\chi| > |J'|$ will bring the system into a state dominated by the spin chirality term. This could potentially lead to new exotic physics such as a skyrmion lattice [50] or chiral spin liquid ground state [11,51].

Naively these results suggest employing a sub-gap driving protocol that exploits the resonant enhancement of $\chi$ for $\hbar\omega \approx U/n$ while simultaneously minimizing electronic interband transitions. However, numerical studies have shown that for driving frequencies close to the multi-photon resonances the system heats immediately and the spin description becomes invalid [11]. In addition, since the real Mott gap is not at $U$ but at the slightly smaller value $U - xt$ (with $x$ a numerical factor of order unity) [52], the frequency has to be chosen below this gap to avoid heating. In the following we therefore focus on photon energies $\hbar\omega/U \approx 0.8$, which is below the Mott gap for $x < 5$.

Assuming a realistic field strength $E \approx 10^9$ V/m, $U = 1.25$ eV, $a = 5$ Å and $\hbar\omega \approx 1$ eV, an effective field strength $\lambda \approx 0.5$ can be achieved. Because interband transitions are avoided in this driving protocol, larger field strengths may still yet be applied without inflicting material damage or other detrimental effects that would disrupt the induced scalar spin chirality. However, even for $\lambda \approx 0.5$ it is possible to open a bandgap of magnitude $\Delta \approx 2$ meV (see Fig. 2b). In contrast, a treatment based on the Aharanov-Casher effect in pure spin systems leads to a field strength $\lambda_m = (g\mu_B E a)/(\hbar c^2)$ [27], which is smaller than the electronic equivalent by a factor $\lambda_m/\lambda_e = (g\mu_B \omega)/(ec^2) \approx 10^{-5}$. Since $\chi \sim \lambda^2$ for small $\lambda$, this leads to a reduction of the gap size by about $10^{-10}$ making an experimental realization of topological magnon systems based on the Aharanov-Casher effect highly challenging. In contrast, the driving protocol proposed here opens a topological gap well within reach of experimental probes.

In optical lattices, heating rates have been shown to be manageable even for $\lambda > 1$ [53]. The magnon band gap can hence be enhanced to values above the currently accessible temperature scales [54].

# 7 Validating the Floquet treatment

To validate the Floquet treatment we compare the results obtained via the static Floquet Hamiltonian with numerical results from time-propagating the system with a quasi-periodic spin Hamiltonian (for details see Appendix C). We take the external field to be switched on adiabatically over approximately 350 periods $T = 2\pi/\omega$, after which we propagate the system for an additional 1750 periods.

To visualize the magnon edge states we consider the spectral function $A_{\mathbf{k}}(\epsilon, \tau)$. For a non-equilibrium system the time-dependent spectral function is defined by

$$A_{\mathbf{k}}(\epsilon, \tau) = i \int \frac{d\bar{\tau}}{2\pi} e^{i\epsilon\bar{\tau}} [G_{\mathbf{k}}^> - G_{\mathbf{k}}^<](\tau + \frac{\bar{\tau}}{2}, \tau - \frac{\bar{\tau}}{2}). \tag{7}$$

The lesser Green's function is proportional to the distribution function $f$ of the initial state, and therefore $G_{\mathbf{k}}^<(\tau, \tau') = 0$ since we start the time-evolution from the magnon ground state. The greater Green's function is given by

$$G_{\mathbf{k}}^>(\tau, \tau') = -i \sum_s \mathrm{Tr}\left(|s\mathbf{k}(\tau)\rangle \langle s\mathbf{k}(\tau')|\right), \tag{8}$$

---

The values of $t$ and $J_D$ were then chosen by comparison to the equilibrium values of $J$ and $J'$ reported in Ref. [28].

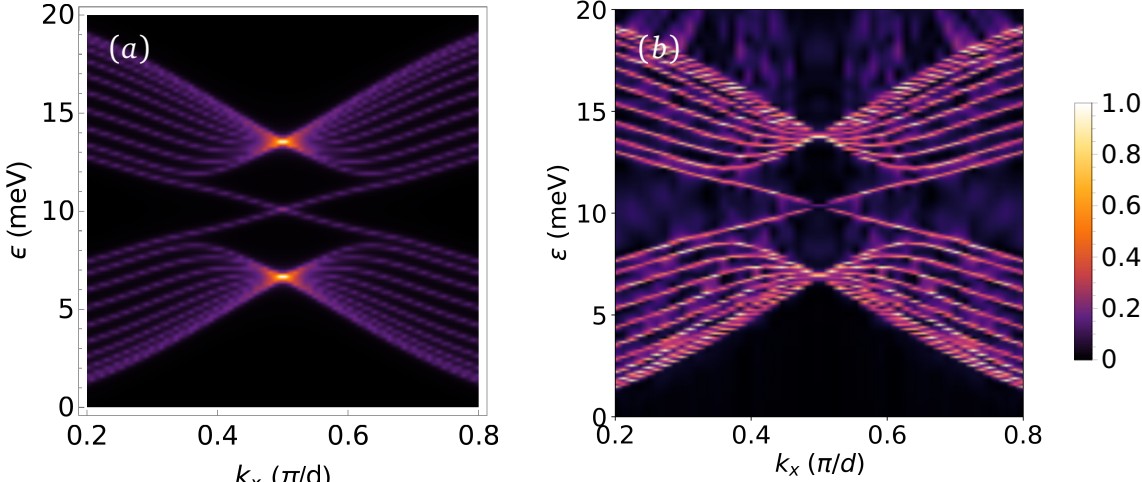

Figure 3: **Spectral functions for a ribbon geometry.** (a) Floquet spectral function $A_{\mathbf{k}}^F(\epsilon)$ for a ribbon with $n_y = 20$ and spin parameters $S = 3/2$, $J = 2.26$ meV, $J' = 0.05$ meV and $\chi = 0.13$ meV. (b) Non-equilibrium spectral function $A_{\mathbf{k}}(t, \epsilon)$ at $\tau = 8.2$ ps for a zig-zag ribbon with $n_y = 20$ and electronic parameters $t = 50$ meV, $U = 1.25$ eV and $J_D = 12$ meV. The optical field has a frequency $\hbar\omega = 1$ eV and field strength $E = 10^9$ V/m. The values of the spectral functions are normalized to the highest value, and the values range from zero (black) to one (white).

where $|s\mathbf{k}(\tau)\rangle = U(\tau)|s\mathbf{k}\rangle$ are the time-evolved single-magnon eigenstates $|s\mathbf{k}\rangle$ of the equilibrium Hamiltonian, $U(\tau) = \mathcal{T}\{e^{-i\int_0^\tau d\bar{\tau}H(\bar{\tau})}\}$ is the time-ordered evolution operator, and the trace is over all single-magnon states. Since $G_{\mathbf{k}}$ is diagonal in $\mathbf{k}$, we can calculate the spectral function by separately time-propagating the states $|s\mathbf{k}\rangle$ for each $\mathbf{k}$. In equilibrium $|s\mathbf{k}(\tau)\rangle = e^{-i\epsilon_{s\mathbf{k}}\tau}|s\mathbf{k}\rangle$, and we find the Floquet spectral function

$$A_{\mathbf{k}}^F(\epsilon) = 2\pi \sum_s \delta(\epsilon - \epsilon_{s\mathbf{k}}). \tag{9}$$

In Fig. 3 we compare the Floquet spectral function $A_{\mathbf{k}}^F(\epsilon)$ and the non-equilibrium spectral function $A_{\mathbf{k}}(\epsilon, \tau)$ for a ribbon with $n_y = 20$. We find a very good agreement between the Floquet and non-equilibrium spectral functions, indicating that for the given parameters the static Floquet Hamiltonian provides a good description of the non-equilibrium magnon dynamics.

## 8 Suggested experimental realizations

We end the paper with a discussion of possible materials and experiments that would support the presence of a topological magnon band structure in a driven system. We note that so far, there has been no experiments that address topological magnons in a non-equilibrium setting. However, equilibrium studies of ferromagnetic bulk $CrI_3$ and antiferromagnetic $Cu_3TeO_6$ have found a magnon band structure consistent with a non-trivial topology attributed to either next-nearest neighbor Dzyaloshinskii-Moriya interactions [28], nearest neighbor Kitaev interactions [55], or the lattice structure [56].

Since our effective Hamiltonian was derived for $S = 1/2$, it gives a simplified description of $S = 3/2$ ferromagnets such as $CrI_3$. However, similar spin Hamiltonians have been used to successfully describe the magnon excitations in $CrI_3$ [28,57]. The main effects of including the $t_{2g}$ orbitals of $Cr^{3+}$ via a Kanamori-Hubbard model (except for an obvious renormalization

of the spin parameters), are spin-orbit coupling and the appearance of biquadratic exchange terms $(\hat{\mathbf{S}}_i \cdot \hat{\mathbf{S}}_j)^2$ [58,59]. Biquadratic exchange can generate nematic instabilities [60,61] and break the $C_6$ rotation symmetry down to the $C_3$ subgroup, generating a trivial mass term that competes with Haldane mass generated from the breaking of time-reversal symmetry and can trivialize the magnon band topology. To estimate this effect, we performed density functional theory calculations in the DFT+$U$ formalism with the Octopus code [62,63] to estimate the size of the trivial mass term in monolayer CrI$_3$ (with the value of $U$ self-consistently determined via the ACBN0 hybrid functional [64]). We find a ground state with $C_6$ symmetry to a numerical accuracy $10^{-6}$, and thus conclude that the effects of the biquadratic terms on the topology of the magnon bands should be negligible in CrI$_3$. This indicates that CrI$_3$ is a promising candidate to study photo-induced topological magnons, although more tailored studies are needed to verify this claim.

Ultracold fermions in optical lattices naturally realize the Hubbard Hamiltonian, and $V$ and $J_D$ are typically negligible [65]. Nevertheless, ferromagnetic spin models can be implemented using near-resonant periodic driving [24]. For most systems, $S = 1/2$ (where we expect our results to still hold approximately), but magnetic correlations for larger $S$ have also been observed using alkali-earth-like atoms [66].

As shown above, a non-zero scalar spin chirality leads to a topological gap, and can generally be probed by Faraday or Kerr rotation measurements [22]. The associated gap opening at the **K**-point in the magnon dispersion will affect the two-magnon optical excitation spectra, as probed by THz spectroscopy [67] or Raman and Brillouin spectroscopy [68]. However, the details of the optical spectra in these types of experiments would require dedicated calculations. The magnon edge states could potentially be probed directly using non-local magnon transport techniques, where magnons can be (detected) injected via the (inverse) spin Hall effect in platinum strips [69]. Finally, resonant inelastic X-ray scattering can be used to probe the magnon dispersion [70].

In optical lattices, a spectroscopic probe could be implemented using oscillating magnetic field gradients. In addition, static gradients can be used to imprint magnons with specific wavenumbers. Their subsequent dynamics gives access to the magnon dispersion and can be probed using spin- and site-resolved detection [71].

# 9 Conclusions

To summarize, we have demonstrated that non-equilibrium driving based on periodic laser fields coupling to charge degrees of freedom can induce topological magnon edge states in the spin sector of prototypical two-dimensional quantum magnets. Specifically, for recently discovered monolayer van der Waals magnets such as CrI$_3$, we predict that a scalar spin chirality term can be induced leading to a sizeable magnon bandgap under realistic driving conditions. This opens the door for potential all-optical topological spintronics applications.

However, an important open problem for future studies is the question of how magnon edge states can be populated in a controlled fashion. Here we note that the situation is different compared to optical engineering of electronic systems, where the generation of dressed Floquet bands, their population, as well as the associated material heating, are intimately linked [20]. In the present work the separation between photon and magnon energy scales means that driving does not automatically populate the magnon bands, and as discussed above heating is largely avoided by adopting a sub-gap driving protocol. Populating the magnon states thus becomes a separate issue to be dealt with in addition to the generation of the non-trivial Floquet bands. We note that population by direct optical pumping have been discussed and experimentally verified for chiral edge states in topological exciton-polariton systems [72–76].

However, direct optical population of chiral magnon edge states through dipolar excitation is usually not possible and one should rather explore indirect mechanisms, for instance through two-magnon Raman scattering.

As an alternative route towards engineering topological magnon edge states with light, non-classical photon fields in cavities can be employed to control magnetic exchange interactions [77,78] and induce nontrivial topology with chiral light modes [79,80]. We also envisage the possibility to combine optical engineering with the control offered by bilayer Moiré systems [81] to induce and control topological magnons.

## Acknowledgments

We acknowledge inspiring discussions with Abhisek Kole, Jin Zhang, Lede Xian and Claudio Verdozzi. We acknowledge support by the Max Planck Institute - New York City Center for Non-Equilibrium Quantum Phenomena. MAS acknowledges support by the DFG through the Emmy Noether programme (SE 2558/2-1). This work was supported by the European Research Council (ERC-2015-AdG694097), the Cluster of Excellence "Advanced Imaging of Matter" (AIM), and Grupos Consolidados (IT1249-19). The Flatiron Institute is a Division of the Simons Foundation.

## A  Details on the effective spin Hamiltonian

We here provide some additional details on the derivation of the effective spin Hamiltonian used in the main text. Assuming that the electronic system is at half-filling and that $U \gg t$, doubly occupied sites will be strongly penalized and the effective Hilbert space is defined by projecting the full Hilbert space onto the subspace of states without doubly occupied sites. For virtual excitations out of the low-energy subspace, where exactly one doubly occupied site is involved, we can rewrite the Hamiltonian up to an irrelevant constant as [37]

$$H_1 = -t \sum_{\langle ij \rangle \sigma} e^{i\theta_{ij}(\tau)} c^\dagger_{i\sigma} c_{j\sigma} + U \sum_i \hat{n}_{i\uparrow} \hat{n}_{i\downarrow} - J_D \sum_{\langle ij \rangle} \hat{\mathbf{S}}_i \cdot \hat{\mathbf{S}}_j , \tag{10}$$

where $U = U_0 - V$. For a circularly polarized laser with a slowly varying envelope $E(\tau)$, the Hamiltonian is almost periodic with the period $H(\tau + 2\pi/\omega) = H(\tau)$ and the Peierls phases are given by $\theta_{ij}(\tau) = -\lambda \sin(\omega \tau - \zeta \phi_{ij})$. Here $\zeta = \pm 1$ for right/left-handed polarization and the dimensionless quantity $\lambda = eEa/\hbar \omega$ determines the effective field strength of the laser. We can then rewrite the electronic Hamiltonian exactly using a Fourier expansion as

$$H = -t \sum_{\langle ij \rangle \sigma} \sum_{mm'} \mathcal{J}_{m-m'}(\lambda) e^{i(m-m')\zeta \phi_{ij}} c^\dagger_{i\sigma} c_{j\sigma} \otimes |m\rangle \langle m'| + H_I \otimes \mathbf{1} - \sum_m m\omega \otimes |m\rangle \langle m|, \tag{11}$$

expressed in the product space of the electronic Hamiltonian and the space of periodic functions [36] denoted by Fourier modes $|m\rangle$, which can be identified with the classical limit of $m$ absorbed or emitted virtual photons. Here, $\mathcal{J}_m(x)$ is the Bessel function of the first kind of order $m$ and the interaction Hamiltonian is $H_I = U \sum_i \hat{n}_{i\uparrow} \hat{n}_{i\downarrow} - J_D \sum_{\langle ij \rangle} \hat{\mathbf{S}}_i \cdot \hat{\mathbf{S}}_j$.

Using quasi-degenerate perturbation theory to simultaneously integrate out the doubly occupied states and the $m \neq 0$ Floquet states [11,38], the effective honeycomb lattice spin Hamiltonian corresponding to the electronic system is given to fourth order in $t/U$ by

$$\mathcal{H} = \sum_{\langle ij \rangle} J \hat{\mathbf{S}}_i \cdot \hat{\mathbf{S}}_j + \sum_{\langle\langle ik \rangle\rangle} J' \hat{\mathbf{S}}_i \cdot \hat{\mathbf{S}}_k + \sum_{\langle\langle ik \rangle\rangle} \nu_{ik} \chi \hat{\mathbf{S}}_j \cdot (\hat{\mathbf{S}}_i \times \hat{\mathbf{S}}_k).$$

Here $J$ and $J'$ are respectively the nearest and next-nearest neighbor light-induced Heisenberg exchanges, and $\chi$ is the synthetic scalar spin chirality. The full expressions for the spin parameters are given by

$$J = -J_D + \Lambda^{(0)} + \frac{1}{2}\sum_{\mathbf{m}}\left(\Lambda_{\mathbf{m}}^{(1)}\cos\left[\frac{\pi(m_1-m_3)}{3}\right] + \Lambda_{\mathbf{m}}^{(1)}\cos\left[\frac{\pi(m_1-m_2+m_3)}{3}\right]\right. \tag{12a}$$
$$\left. -\frac{1}{2}\Lambda_{\mathbf{m}}^{(2)}\cos\left[\frac{\pi m_2}{3}\right] - 2\Lambda_{\mathbf{m}}^{(2)}\cos[\pi m_2]) - 3\Gamma_{\mathbf{m}}\right)$$

$$J' = -\frac{1}{2}\sum_{\mathbf{m}}\left(\Lambda_{\mathbf{m}}^{(1)}\cos\left[\frac{\pi(m_1-m_2+m_3)}{3}\right] - \Lambda_{\mathbf{m}}^{(2)}\cos\left[\frac{\pi m_2}{3}\right] - 2\Gamma_{\mathbf{m}}\right) \tag{12b}$$

$$\chi = \sum_{\mathbf{m}}\left(\Lambda_{\mathbf{m}}^{(1)}\sin\left[\frac{\pi(m_1-m_2+m_3)}{3}\right] - \Lambda_{\mathbf{m}}^{(2)}\sin\left[\frac{\pi m_2}{3}\right]\right), \tag{12c}$$

where $\mathbf{m} = \{m_1, m_2, m_3\}$ are Floquet indexes counting how many virtual photons have been absorbed or emitted and

$$\Lambda^{(0)} = 4t^2\sum_m \frac{\mathcal{J}_m(\lambda)^2}{U - m\omega} \tag{13a}$$

$$\Lambda_{\mathbf{m}}^{(1)} = 8t^4\frac{\mathcal{J}_{m_1}(\lambda)\mathcal{J}_{m_2-m_1}(\lambda)\mathcal{J}_{m_2-m_3}(\lambda)\mathcal{J}_{m_3}(\lambda)}{(U-m_1\omega)(U-m_2\omega)(U-m_3\omega)} \tag{13b}$$

$$\Lambda_{\mathbf{m}}^{(2)} = 16t^4(1-\delta_{m_2,0})(-1)^{m_1-m_3}\cos^2\left[\frac{\pi m_2}{2}\right]\frac{\mathcal{J}_{m_1}(\lambda)\mathcal{J}_{m_2-m_1}(\lambda)\mathcal{J}_{m_2-m_3}(\lambda)\mathcal{J}_{m_3}(\lambda)}{m_2\omega(U-m_1\omega)(U-m_3\omega)} \tag{13c}$$

$$\Gamma_{\mathbf{m}} = 4t^4\left[\frac{\delta_{m_3,0}}{U-m_1\omega} + \frac{\delta_{m_3,0}}{U-m_2\omega}\right]\frac{\mathcal{J}_{m_1}^2(\lambda)\mathcal{J}_{m_2}^2(\lambda)}{(U-m_1\omega)(U-m_2\omega)}. \tag{13d}$$

## B  Details on the processes inducing a scalar spin chirality

After integrating out the doubly occcupied states and the Floquet states with non-zero index $m$, the fourth order contributions to the spin Hamiltonian separate into three physically distinct processes, depending on the state at the second intermediate step of the virtual process. In the first class of processes the second intermediate state contains a single doublon-holon pair and non-zero Floquet index, as described by terms of the form

$$H^{(1)} = -\sum_{ijkl}\sum_{\substack{\sigma_1\sigma_2\\\sigma_3\sigma_4}}\sum_{\substack{m_1m_2\\m_3}}\left[c_{i\sigma_1}^\dagger c_{j\sigma_1}c_{j\sigma_2}^\dagger c_{k\sigma_2}c_{k\sigma_3}^\dagger c_{l\sigma_3}c_{l\sigma_4}^\dagger c_{i\sigma_4}\frac{t_{ij}^{-m_3}t_{jk}^{m_3-m_2}t_{kl}^{m_2-m_1}t_{li}^{m_1}}{(U-m_1\omega)(U-m_2\omega)(U-m_3\omega)}\right.$$

$$\tag{14}$$

$$+c_{j\sigma_2}^\dagger c_{k\sigma_2}c_{i\sigma_1}^\dagger c_{j\sigma_1}c_{k\sigma_3}^\dagger c_{l\sigma_3}c_{l\sigma_4}^\dagger c_{i\sigma_4}\frac{t_{jk}^{-m_3}t_{ij}^{m_3-m_2}t_{kl}^{m_2-m_1}t_{li}^{m_1}}{(U-m_1\omega)(U-m_2\omega)(U-m_3\omega)}$$

$$+c_{j\sigma_2}^\dagger c_{k\sigma_2}c_{k\sigma_3}^\dagger c_{l\sigma_3}c_{i\sigma_1}^\dagger c_{j\sigma_1}c_{l\sigma_4}^\dagger c_{i\sigma_4}\frac{t_{jk}^{-m_3}t_{kl}^{m_3-m_2}t_{ij}^{m_2-m_1}t_{li}^{m_1}}{(U-m_1\omega)(U-m_2\omega)(U-m_3\omega)}$$

$$\left.+c_{k\sigma_3}^\dagger c_{l\sigma_3}c_{j\sigma_2}^\dagger c_{k\sigma_2}c_{i\sigma_1}^\dagger c_{j\sigma_1}c_{l\sigma_4}^\dagger c_{i\sigma_4}\frac{t_{kl}^{-m_3}t_{jk}^{m_3-m_2}t_{ij}^{m_2-m_1}t_{li}^{m_1}}{(U-m_1\omega)(U-m_2\omega)(U-m_3\omega)}\right].$$

In the second class of processes the second intermediate state returns to half-filling at non-zero Floquet index, as described by terms of the form

$$
\begin{aligned}
H^{(2)} = -\sum_{ijl} \sum_{\substack{\sigma_1\sigma_2 \\ \sigma_3\sigma_4}} \sum_{\substack{m_1 m_2 \\ m_3}} \Bigg[ & c_{i\sigma_1}^\dagger c_{j\sigma_1} c_{j\sigma_2}^\dagger c_{i\sigma_2} c_{i\sigma_3}^\dagger c_{l\sigma_3} c_{l\sigma_4}^\dagger c_{i\sigma_4} \frac{(1-\delta_{m_2,0}) t_{ij}^{-m_3} t_{ji}^{m_3-m_2} t_{il}^{m_2-m_1} t_{li}^{m_1}}{(U-m_1\omega)(m_2\omega)(U-m_3\omega)} \\
& + c_{j\sigma_2}^\dagger c_{i\sigma_2} c_{i\sigma_1}^\dagger c_{j\sigma_1} c_{i\sigma_3}^\dagger c_{l\sigma_3} c_{l\sigma_4}^\dagger c_{i\sigma_4} \frac{(1-\delta_{m_2,0}) t_{ji}^{-m_3} t_{ij}^{m_3-m_2} t_{il}^{m_2-m_1} t_{li}^{m_1}}{(U-m_1\omega)(m_2\omega)(U-m_3\omega)} \Bigg] \\
-\sum_{ijk} \sum_{\substack{\sigma_1\sigma_2 \\ \sigma_3\sigma_4}} \sum_{\substack{m_1 m_2 \\ m_3}} \Bigg[ & c_{j\sigma_2}^\dagger c_{k\sigma_2} c_{k\sigma_3}^\dagger c_{j\sigma_3} c_{i\sigma_1}^\dagger c_{j\sigma_1} c_{j\sigma_4}^\dagger c_{i\sigma_4} \frac{(1-\delta_{m_2,0}) t_{jk}^{-m_3} t_{kj}^{m_3-m_2} t_{ij}^{m_2-m_1} t_{ji}^{m_1}}{(U-m_1\omega)(m_2\omega)(U-m_3\omega)} \\
& + c_{k\sigma_3}^\dagger c_{j\sigma_3} c_{j\sigma_2}^\dagger c_{k\sigma_2} c_{i\sigma_1}^\dagger c_{j\sigma_1} c_{j\sigma_4}^\dagger c_{i\sigma_4} \frac{(1-\delta_{m_2,0}) t_{kj}^{-m_3} t_{jk}^{m_3-m_2} t_{ij}^{m_2-m_1} t_{ji}^{m_1}}{(U-m_1\omega)(m_2\omega)(U-m_3\omega)} \Bigg].
\end{aligned}
\tag{15}
$$

In the third class of processes the second intermediate state returns to half-filling at zero Floquet index, as described by terms of the form

$$
\begin{aligned}
H^{(3)} = \frac{1}{2} \sum_{ijl} \sum_{\substack{\sigma_1\sigma_2 \\ \sigma_3\sigma_4}} \sum_{m_1 m_2} \Bigg[ & c_{k\sigma_1}^\dagger c_{l\sigma_1} c_{l\sigma_2}^\dagger c_{k\sigma_2} c_{i\sigma_3}^\dagger c_{j\sigma_3} c_{j\sigma_4}^\dagger c_{i\sigma_4} t_{kl}^{-m_2} t_{lk}^{m_2} t_{ij}^{-m_1} t_{ji}^{m_1} \\
& \times \left( \frac{1}{(U-m_1\omega)(U-m_2\omega)^2} + \frac{1}{(U-m_1\omega)^2(U-m_2\omega)} \right) \Bigg].
\end{aligned}
\tag{16}
$$

In order for a process to induce a non-zero scalar spin chirality the electrons need to accumulate a net phase in the light field. Equivalently, the sites involved in the process needs to enclose a non-zero area. Such processes lead to time-reversal symmetry breaking in analogy with the net phase accumulation of electrons moving in closed loops in an external magnetic field, however with the difference that driving with a circularly polarized electric field conserves the full $SU(2)$ spin symmetry.

On the honeycomb lattice, a scalar spin chirality arises due hopping processes that enclose an isosceles triangle within the hexagons, as indicated schematically in Fig. 4. Here the pink and blue lines encircle the sites involved in a process, and the orange dot shows the site at which the system returns to half-filling in the second intermediate step. The gray areas show the enclosed area corresponding in the non-zero cases to an isosceles triangle. We note that processes in the third class, described by the Hamiltonian $H^{(3)}$ above, can be viewed as two separate second-order processes and so does not lead to a net spin chirality. Also shown in Fig. 4 is a process in the first class, containing a single doublon-holon pair in the second intermediate state.

## C   Details on the time-evolution and non-equilibrium spectral functions

In the main text we validate the Floquet treatment based on the effective spin Hamiltonian by a comparison to numerical results obtained via time-propagating the system with a quasi-periodic spin Hamiltonian. We take the external electric field as

$$
\mathbf{E}(\tau) = E \sin\left( \frac{\pi\tau}{2\tau_0} \right) (\cos\omega\tau, \zeta\sin\omega\tau)(\theta(\tau) - \theta(\tau-\tau_0)) + E(\cos\omega\tau, \zeta\sin\omega\tau)\theta(\tau-\tau_0),
\tag{17}
$$

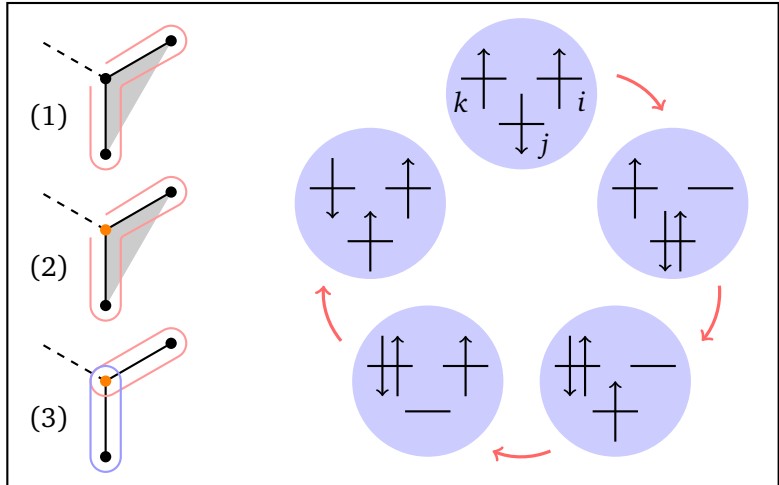

Figure 4: **Light-induced scalar spin chirality.** Illustration of the three classes of fourth-order hopping processes possible on the honeycomb lattice: (1) a process where the second intermediate state contains a single holon-doublon pair at non-zero Floquet index. (2) A process where the system returns to half-filling in the second intermediate step (at the orange site) at non-zero Floquet index. (3) A process where the system returns to half-filling and zero Floquet index in the second intermediate step (at the site indicated in orange). The right panel gives an example of a process in class (1) that breaks time-reversal symmetry and induces a scalar spin chirality.

where the envelope provides an adiabatic switch-on of the external field and $\tau_0$ is assumed to be so large that the envelope can be taken as constant over the period of the field ($\tau_0 \gg 2\pi/\omega$). We can then approximate the Peierls phases as $\theta_{ij}(\tau) \approx -\lambda(\tau)\sin(\omega\tau - \zeta\phi_{ij})$ where $\lambda(\tau) = (eEa/\hbar\omega)\sin(\pi\tau/2\tau_0)$. The dynamic spin Hamiltonian is of the same form as its static counterpart but with the time-dependent spin parameters

$$J(\tau) = -J_D + \sum_{\mathbf{m}} \Lambda_{\mathbf{m}}^{(0)}(\tau)\cos(2m_4\omega\tau) \tag{18a}$$

$$+ \frac{1}{2}\sum_{\mathbf{m}}\left[2\Lambda_{\mathbf{m}}^{(0)}(\tau) + \Lambda_{\mathbf{m}}^{(1)}(\tau)\cos\left(\frac{\pi(m_1-m_3)}{3} + m_4\omega\tau\right)\right.$$

$$+ \Lambda_{\mathbf{m}}^{(1)}(\tau)\cos\left(\frac{\pi(m_1-m_2+m_3)}{3} + m_4\omega\tau\right)$$

$$\left. - \frac{1}{2}\Lambda_{\mathbf{m}}^{(2)}(\tau)\cos\left(\frac{\pi m_2}{3} + m_4\omega\tau\right) - 2\Lambda_{\mathbf{m}}^{(2)}(\tau)\cos(\pi m_2 + m_4\omega\tau) - 3\Gamma_{\mathbf{m}}(\tau)\right]$$

$$J'(\tau) = -\frac{1}{2}\sum_{\mathbf{m}}\left[\Lambda_{\mathbf{m}}^{(1)}(\tau)\cos\left(\frac{\pi(m_1-m_2+m_3)}{3} + m_4\omega\tau\right) - \Lambda_{\mathbf{m}}^{(2)}(\tau)\cos\left(\frac{\pi m_2}{3} + m_4\omega\tau\right)\right.$$

$$\left. - 2\Gamma_{\mathbf{m}}(\tau)\right] \tag{18b}$$

$$\chi(\tau) = \sum_{\mathbf{m}}\left[\Lambda_{\mathbf{m}}^{(1)}(\tau)\sin\left(\frac{\pi(m_1-m_2+m_3)}{3} + m_4\omega\tau\right) - \Lambda_{\mathbf{m}}^{(2)}(\tau)\sin\left(\frac{\pi m_2}{3} + m_4\omega\tau\right)\right]. \tag{18c}$$

Here $\mathbf{m} = \{m_1, m_2, m_3, m_4\}$ gives a sum over four Floquet indexes, and the parameters $\Lambda_{\mathbf{m}}^{(n)}$ and $\Gamma_{\mathbf{m}}$ are given by

$$\Lambda_{\mathbf{m}}^{(0)}(\tau) = 4t^2\delta_{m_2,0}\delta_{m_3,0}\frac{\mathcal{J}_{m_1}(\lambda)\mathcal{J}_{m_1-2m_4}(\lambda)}{U - m_1\omega} \tag{19a}$$

$$\Lambda_{\mathbf{m}}^{(1)}(\tau) = 8t^4 \frac{\mathcal{J}_{m_1}(\lambda)\mathcal{J}_{m_2-m_1}(\lambda)\mathcal{J}_{m_2-m_3}(\lambda)\mathcal{J}_{m_3-m_4}(\lambda)}{(U-m_1\omega)(U-m_2\omega)(U-m_3\omega)} \tag{19b}$$

$$\Lambda_{\mathbf{m}}^{(2)}(\tau) = 16t^4(1-\delta_{m_2,0})(-1)^{m_1-m_3}\cos^2\left[\frac{\pi m_2}{2}\right]\frac{\mathcal{J}_{m_1}(\lambda)\mathcal{J}_{m_2-m_1}(\lambda)\mathcal{J}_{m_2-m_3}(\lambda)\mathcal{J}_{m_3-m_4}(\lambda)}{(U-m_1\omega)(m_2\omega)(U-m_3\omega)} \tag{19c}$$

$$\Gamma_{\mathbf{m}}(\tau) = 4t^4\left[\frac{\delta_{m_3,0}}{U-m_1\omega} + \frac{\delta_{m_3,0}}{U-m_2\omega}\right]\frac{\mathcal{J}_{m_1}^2(\lambda)\mathcal{J}_{m_2}(\lambda)\mathcal{J}_{m_2-m_4}(\lambda)}{(U-m_1\omega)(U-m_2\omega)}. \tag{19d}$$

The time-dependence of these parameters is given implicitly by the dependence on $\lambda(\tau)$, but for the sake of readability we have suppressed the time dependence of $\lambda$ in the equations above.

Writing the Hamiltonian in the spin-wave approximation we evaluate the time-dependent spectral function

$$A_{\mathbf{k}}(\epsilon, \tau) = i\int \frac{d\bar{\tau}}{2\pi}e^{i\epsilon\bar{\tau}}[G_{\mathbf{k}}^> - G_{\mathbf{k}}^<](\tau + \frac{\bar{\tau}}{2}, \tau - \frac{\bar{\tau}}{2}). \tag{20}$$

As noted in the main text the lesser Green's function is proportional to the distribution function $f$ of the initial state, and therefore vanishes if we start the time-evolution from the magnon ground state. The greater Green's function is given by

$$G_{\mathbf{k}}^>(\tau, \tau') = -i\sum_s \mathrm{Tr}\left(|s\mathbf{k}(\tau)\rangle\langle s\mathbf{k}(\tau')|\right), \tag{21}$$

where $|s\mathbf{k}(\tau)\rangle = U(\tau)|s\mathbf{k}\rangle$ are the time-evolved single-magnon eigenstates $|s\mathbf{k}\rangle$ of the equilibrium Hamiltonian, $U(\tau) = \mathcal{T}\{e^{-i\int_0^\tau d\bar{\tau}H(\bar{\tau})}\}$ is the time-ordered evolution operator, and the trace is over all single-magnon states. Since $G_{\mathbf{k}}$ is diagonal in $\mathbf{k}$, we can calculate the spectral function by separately time-propagating the states $|s\mathbf{k}\rangle$ for each $\mathbf{k}$.

For the results presented in the main text we considered a zig-zag ribbon with $n_y = 20$ and electronic parameters $t = 50$ meV, $U = 1.25$ eV and $J_D = 12$ meV. The electric field has a frequency $\hbar\omega = 1$ eV and field strength $E = 10^9$ V/m, and was switched on over a time $\tau_0 = 1.3$ ps.

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
