# Peer review of "Light-induced topological magnons in two-dimensional van der Waals magnets"

_SciPost Physics, doi:SciPost Phys. 9, 061 (2020)_

## Round 2 · Referee Report · Anonymous (Referee 1) · 2020-9-17

Report

This manuscript theoretically investigates a possibility of inducing topological magnons in honeycomb ferromagnets irradiated by a circularly-polarised light. The effective model for magnons is identified as being analogous to Haldane's, and time evolution of the spectrum and the relevant light-matter coupling are studied. Given the recent upsurge of general interests in non-equilibrium physics, and Floquet engineering in particular, the subject of the present work is certainly interesting, especially because studies on Floquet physics for magnets are still scarce as compared with those for electron systems. The paper is well written and the results seem to be sound. However, I have some reservations as follows.

  1. The authors stress, already in the abstract, that their scheme should apply to CrI3. The material certainly is a honeycomb magnet, but a brief section ("Justifying ...") on CrI3 is not detailed or convincing enough for the material application. Specifically, CrI3 is a S=3/2 magnet, while the authors state that their S=1/2 model "gives a simplified description", but they should explain this in more detail. The captions for Figs.1(c) and 3 also state that S=3/2, but proper explanations are not given in the text. In some cases you can decompose S=3/2 system into S=1/2 sectors, but there can be a lot of subtleties, especially when one wants to discuss Haldane physics.

  2. As for the magnon dynamics, it is well-known that the equation of motion for magnons in magnets has the same form as the Schroedinger's equation for electron systems in the tight-binding model. In this sense, the fact that we end up with Haldane's model for the magnons in a honeycomb lattice in a circularly-polarised light (CPL) is understandable as a magnetic analogue of the Floquet topological insulator in the electrons in a honeycomb lattice in a CPL. It is of course highly nontrivial how the actual Floquet spin Hamiltonian is obtained, which is precisely the subject of Claassen et al [11] and Kitamura et al [20]. Still, in the context of the Floquet topological insulator, the original proposal for by Oka and Aoki [Phys. Rev. B 79, 081406(R) (2009)] may be cited.

  3. The authors emphasise that their light-matter coupling is some 10^5 times larger than in the "similar scheme" of the Aharonov-Casher effect, in the abstract and the text, but the latter is a totally different phenomenon, so it might be confusing to directly compare them.

Some minor points: Fig.3: A colour bar should be attached.

Thus I conclude that the manuscript should be amended before it is accepted for publication in this Journal.

  • validity: -
  • significance: -
  • originality: -
  • clarity: -
  • formatting: -
  • grammar: -

Author:  Emil Vinas Boström  on 2020-10-23  [id 1016]

(in reply to Report 1 on 2020-09-17)

  1. We agree with the Referee that the model considered in the manuscript is too simplistic to describe the detailed magnetic properties of CrI$_3$. However, we believe it still bears merits as a starting point for more detailed studies of light-induced magnetic phenomena in magnetic vdW materials. We have therefore, in the revised manuscript, shifted the main focus to light-induced topological magnons in generic $S = 1/2$ honeycomb ferromagnets, and included a discussion of CrI$_3$ as a possible material realization (pointing out some of the subtleties in extending the model to $S = 3/2$ systems).

  2. We agree with the Referee that there is a strong analogy between the properties of the electronic and bosonic Haldane models, and how they may be realized out of equilibrium. In the revised manuscript, we have included a citation to the original work by Oka and Aoki.

  3. Although we agree with the Referee that the light-matter coupling considered in the manuscript is of a different physical origin than a coupling based on the Aharanov-Casher effect, we believe a direct comparison between the two is both possible and well defined. Therefore, we here elaborate on how this comparison is made and what it implies. In both approaches, driving the system leads to additional phase factors for interactions along on a bond, whose strength can be quantified by a dimensionless coupling constant $\lambda$. Moreover, the value of $\lambda$ is what controls the photo-induced renormalization of the spin parameters, as it appears as the argument of the Bessel functions defining the non-equilibrium spin parameters. In the manuscript we show that our light-matter coupling leads to a value $\lambda_e = (eEa)/(\hbar\omega)$, while the corresponding value arising from an Aharanov-Casher coupling is $\lambda_m = (g\mu_B Ea)/(\hbar c^2)$ [Owerre, Journal of Physics Communications 1(2), 021002 (2017)]. It is then straightforward to verify that for optical wavelengths and a laser field strengths of $E \approx 10^9$ V/m, the ratio of the two couplings $\lambda_m/\lambda_e = (g\mu_B \omega)/(ec^2) \approx 10^{-5}$. The value of this ratio is not only of academic interest, but has important physical consequences in particular for the magnitude of the photo-induced topological gap. Since even for strong lasers fields we find $\lambda < 0.1$, the Bessel functions defining the spin parameters can be expanded to show that the magnon band gap $\Delta \sim \lambda^2$. Thus, compared to a driving scheme exploiting the Aharanov-Casher effect, the band gap we find is enhanced by a factor $10^{10}$. This enhancement will be of crucial importance for any experimental realization.

Minor points: The values shown in the spectral functions of Fig. 3 are normalized to the highest value, which is explained more clearly in the figure caption of the revised manuscript. We've also inserted a color bar for clarity.

---

## Round 2 · Referee Report · Anonymous (Referee 2) · 2020-9-28

Report

In the present manuscript, the authors have investigated the effect of a periodic laser pulse on 2D hexagonal lattices using the Floquet formalism. The main finding is light-induced three-spins chiral interaction in the system. This chiral interaction opens a topological gap in the band structure of magnons leads to a phase transition to topological magnon phase.
I believe, the manuscript is interesting, and the finding is important. But before any final decision I want to ask the author to address the following questions and comments:
1- The effective Floquet Hamiltonian, Eq. 4, was written up to the next-nearest-neighbor (NNN). To be consistent, I think the tight-binding Hamiltonian, Eq. 1, should also be written up to NNN. This means that three terms should be added: 1. intrinsic spin-orbit interaction (Kane-Mele model), 2. NNN hoping term, and 3. Equilibrium three-spins chiral interaction. The first and third interactions are chiral and compete with the induced three-spins scaler chiral interaction. Please note that in bipartite systems in the presence of NNN, as well as nonbipartite 2D systems, there can be a finite three-spins chiral interaction. The amplitude of this term in a bipartite system is proportional to t_1^2*t_2/U^2, where t_{1(2)} is NN (NNN) hoping amplitude, c.f. the laser-induced chiral term, found by authors, t_1^4/U^3. I am wondering if authors can justify their assumptions to ignore these terms.
2- In a recent study it has been shown that even in the absence of SOI a DMI-like term can be induced by periodic laser pulses, PRL 116, 125301 (2016). I am wondering why this term is not appeared in the Floquet spin Hamiltonian of present study.
3- The authors claim, “the effects of the biquadratic terms are negligible in CrI3”. I do not think this is correct and this is still an open question. For example, a recent study shows that non-Heisenberg terms are important in CrI_3, see arXiv:2007.14518. In general, I think it is better if the authors do not link their study to a special material like CrI_3 and instead just consider their study as a toy model for generic 2D magnetic insulators with hexagonal lattice structures.
4- Is there any difference between zig-zag and armchair nanoribbons in this system?
5- Several notations are confusing: “t” has been used for both time and NN hoping. The density operator appeared in NN Coulomb interaction in Eq.1 and the footnote 1 should have a spin index or if the authors use another notation, they should write it explicitly. What is index “s” in Eq 5?
6- Can authors add a reference for the Mott gap=U - x*t_1?
7- I think the following papers are relevant to the present study and if applicable the authors should compare their results with them:
a. PRL 115, 075301 (2015).
b. SciPost Phys. 6, 027 (2019).
c. PRB 100, 060410(R) (2019).
d. EPL, 126, 27002 (2019).
e. arXiv:1812.05101.
  • validity: high
  • significance: high
  • originality: high
  • clarity: good
  • formatting: good
  • grammar: good

Author:  Emil Vinas Boström  on 2020-10-23  [id 1015]

(in reply to Report 2 on 2020-09-28)

  1. We thank the Referee for this comment, and in the following motivate our choice of model. In the revised manuscript, we focus on the leading-order light-induced phenomena in generic honeycomb ferromagnets. Both next-nearest neighbor and spin-orbit interactions are straightforward to add, but the former will not affect the topology of the magnon bands. Since our system is inversion symmetric this also rules out most spin-orbit interactions, the Kane-Mele interaction being an important exception. To our knowledge there are no estimates of the magnitude of such an interaction in magnetic vdW materials, and we therefore neglect it as a first approximation. Under the assumption that spin-orbit interactions are negligible, the equilibrium scalar spin chirality interaction will vanish since it is proportional to the imaginary part of $t_1^2t_2/U^2$ [Kitamura, Oka and Aoki, Phys. Rev. B 96, 014406 (2017)]. We would like to note that our results hold irrespective of additional terms being present or not. If additional terms are present, and compete with our scalar spin chirality, the respective order of magnitude of these terms can be compared with the one in our manuscript to see which one dominates. Thus, although we agree with the Referee that both spin-orbit interactions and next-nearest neighbor hopping could be of importance in real vdW materials, we believe our work provides an important first step to describe light-induced phenomena in generic honeycomb ferromagnets.

  2. We thank the Referee for spotting this difference, and hope the following explanation is satisfactory: The reason why a DM interaction is found in the work of Bukov, Kolodrubetz and Polkovnikov (BKP) and not in the present work, is due to the different symmetry breakings induced by the driving schemes. In the present case, we consider a circularly polarized laser field coupling to the electron kinetic term via Peierls substitution, which preserves the SU$(2)$ spin symmetry of the system and leads to isotropic spin parameters. In contrast, the BKP paper considers a local Zeeman coupling to the $z$-component of the spin, and employs a field that is different along the $x$- and $y$-directions of the lattice. This term explicitly breaks both the SU$(2)$ spin symmetry and the $C_4$ rotational symmetry of the system. We note that a similar DM interaction as in the BKP paper can be induced by a spin-optical coupling based on the Aharanov-Casher effect [Owerre, Journal of Physics Communications 1(2), 021002 (2017)], where the direction of the DM vector is determined by the anisotropic coupling to the spin operators. In the present case, where the light-matter coupling is isotropic, the DM interaction is replaced by the SU$(2)$ invariant scalar spin chirality term.

  3. We agree with the Referee that biquadratic (and other) terms could be of importance in CrI$_3$, as argued in several recent studies [Chen et al., Phys. Rev. X 8, 041028 (2018), Lee et al., Phys. Rev. Lett. 124, 017201 (2020), Kartsev et al., arXiv:2006.04891]. We have therefore followed the Referee's suggestion and focused the revised manuscript on generic honeycomb ferromagnets. The discussion of CrI$_3$ as a possible material realization (pointing out some of the subtleties in extending the model to $S = 3/2$ systems) has been deferred to the section about possible experimental realizations.

  4. Within our model there is no qualitative difference between the magnons edge states of zig-zag and armchair ribbons, in the sense that their topological properties are unaffected. However, just like in the electronic Haldane model, the ${\bf K}$ and ${\bf K}'$ points of the two-dimensional Brillouin zone are projected onto different momenta of the surface Brillouin zone for a zig-zag ribbon, while they correspond to the same surface momentum for an armchair ribbon. This could have important consequences when trying to populate the magnon states, since this process is typically restricted to small momentum transfers.

  5. We thank the Referee for pointing this out, and have made sure the notation is unique and consistent in the revised manuscript.

  6. We thank the Referee for pointing this out and have added a reference to the original work by Hubbard [Hubbard, Proc. R. Soc. Lond. A 281 401–419, (1964)].

  7. We thank the referee for pointing out these reference, to which we have compared our results and included those we found particularly relevant.

---

## Round 4 · Referee Report · Anonymous (Referee 1) · 2020-11-3

Report

The authors have revised the manuscript following the referee's report basically satisfactorily, so this referee thinks the paper should now be accepted for publication.

---

## Round 4 · Author Response

Dear Editor,

We hereby resubmit our manuscript "Light-induced topological magnons in two-dimensional van der Waals magnets" for consideration in SciPost Physics. We have addressed all the questions and comments raised by the Referees in our responses, and done the appropriate changes to our manuscript. We thank the Referees for helping to improve our manuscript, and hope that our manuscript is now suitable for publication.

Best regards,
Emil Vinas Boström, Martin Claassen, James W. McIver, Gregor Jotzu, Angel Rubio and Michael A. Sentef

---

## Round 4 · List of Changes

We have focused the revised manuscript on light-induced topological magnons in generic S = 1/2 honeycomb ferromagnets, and moved the discussion of CrI3 to the section about possible material realizations.

We have added a brief discussion on the magnon edge states in armchair ribbons.

We have made sure our notation is unique and consistent. In particular, we have used the symbol tau to denote the time variable to reduced the risk of confusion with the hopping parameter t.

We have included a color bar in Fig. 3.

We have added a number of references suggested by the Referees.

---

## Editorial Decision

published